# Visual processing oscillates differently through time for adults with ADHD

Pénélope Pelland-Goulet [1,2,3,4,5☯*], Martin Arguin[1,2☯*], Hélène Brisebois[5,6‡],
Nathalie Gosselin[1,2,3,4‡]

1 Psychology Department, Université de Montréal, Montréal, Québec, Canada, 2 Center for
Interdisciplinary Research on Brain and Learning (CIRCA), Montréal, Québec, Canada, 3 International
Laboratory for Brain, Music and Sound Research (BRAMS), Montréal, Québec, Canada, 4 Center for
Research on Brain, Language and Music (CRBLM), Montréal, Québec, Canada, 5 Centre Alpha-Neuro,
Laval, Québec, Canada, 6 Psychology Department, Collège Montmorency, Laval, Québec, Canada

☯ These authors contributed equally to this work.
‡ HB and NG also contributed equally to this work.
* martin.arguin@umontreal.ca (MA); penelope.pelland-goulet@umontreal.ca (PPG)

University, TAIWAN

**Peer Review History:** PLOS recognizes the
benefits of transparency in the peer review
process; therefore, we enable the publication
of all of the content of peer review and
author responses alongside final, published
articles. The editorial history of this article is
available here: https://doi.org/10.1371/journal.
pone.0310605

## Abstract

ADHD is a neurodevelopmental disorder affecting 3–4% of Canadian adults and
2.6% of adults worldwide. Its symptoms include inattention, hyperactivity and impul-
sivity. Though ADHD is known to affect several brain functions and cognitive pro-
cesses, little is known regarding its impact on perceptual oscillations. This study
compared the temporal features of visual processing between ADHD and neurotypi-
cal individuals in a visual word recognition task through the use of a temporal sam-
pling technique, the outcome of which are classification images reflecting processing
effectiveness according to the temporal properties of the stimulus. These temporal
features were sufficiently different across groups while at the same time sufficiently
congruent across participants of the same group that a machine learning algorithm
classified participants in their respective groups with a 91.8% accuracy using only a
small portion of the available features. Secondary findings showed that individuals
with ADHD could be classified with high accuracy (91.3%) regarding their use of
psychostimulant medication. These findings suggest the existence of strong behav-
ioral markers of ADHD as well as of regular medication usage on visual performance
which can be uncovered by random temporal sampling.

## Introduction

The attention deficit and hyperactivity disorder (ADHD) is a neuropsychological
condition characterised by symptoms of inattention, hyperactivity, and impulsivity
[1] which affects 3–4% of Canadian adults [2,3] and 2.6% of adults worldwide [4].
It has been demonstrated that people with ADHD show functional deficits affecting
sustained attention [5,6], processing speed [7,8], and executive functions [9] such as

**Data availability statement:** The raw dataset in text format and relevant information will be made available on the Borealis repository upon manuscript acceptance. The data can be accessed here: https://doi.org/10.5683/SP3/C5QVH6.

**Funding:** HB & NG : Grant number : 2019-PZ-264929 Fonds de Recherche du Québec - Société et Culture https://frq.gouv.qc.ca/societe-et-culture/ The funders did not play any role in the study design, data collection and analysis, decision to publish or preparation of the manuscript. PPG: Grant number : 752-2023-1085 Social Sciences and Humanities Research Council https://www.sshrc-crsh.gc.ca/funding-financement/programs-programmes/fellowships/doctoral-doctorat-eng.aspx The funders did not play any role in the study design, data collection and analysis, decision to publish or preparation of the manuscript. PPG: Grant number : - Center for Interdisciplinary Research on Brain and Learning, Excellence Fellowship (2024) https://circa.openum.ca/en/financement/opportunites-etudiants/soutien-financier-pour-etudiants-et-spd/ The funders did not play any role in the study design, data collection and analysis, decision to publish or preparation of the manuscript.

**Competing interests:** The authors have declared that no competing interests exist.

working memory [10,11], and inhibition [12]. Persons with ADHD also exhibit several functional cerebral abnormalities which can be related to these cognitive and executive deficits [13].

Another line of investigation for ADHD pertains to cerebral oscillations, which originate from transient neural groups producing repeated synchronized action potentials at a particular frequency. Resting-state EEG studies have shown stronger oscillations in the theta (4–8 Hz) and alpha (8–12 Hz) ranges in ADHD than neurotypical participants ( [14]; see [15] for a review). Relatedly, others report a stronger theta/beta ratio (TBR;13–30 Hz) oscillatory power in ADHD vs neurotypical adults [16–19]. However, other studies of EEG at rest have reported inconsistent findings [15], and objections have been addressed against the literature surrounding the TBR [20,21]. Several investigations have shown that, while carrying out attentional tasks, adults with ADHD exhibit distinct oscillatory patterns that particularly pertain to alpha oscillations [22–25].

Neural oscillations have been argued to constitute the central basis for the functional output of brain activity (e.g. [26]). If this is true, one crucial implication is that this output should be modulated through time. In the case of vision, this would imply temporal variability in processing capacity within a short time scale. A relatively substantial literature on this issue has more or less successfully attempted to demonstrate that visual function oscillates through time at a particular unique frequency or combination thereof, which would be tied to the underlying brain activity (see [27–29] for reviews). Recently, our laboratory has developed a promising technique called random temporal sampling, which offers a strong demonstration of variations of visual processing effectiveness through time and can reveal differences in visual oscillatory mechanisms according to task demands [30], stimulation conditions [31] or the age of participants [32].

The random temporal sampling technique involves the brief (e.g. 200 ms) presentation of stimuli made of an additive combination of signal (the target to be processed) and noise (a patch of visual white noise superimposed on top of the signal), on which a signal-to-noise ratio (SNR) that varies randomly through exposure duration (Fig 1) is added. By separating the temporal samples (i.e. temporal variations of SNR) associated with errors versus correct responses and by subtracting the former from the latter, one obtains a classification image (CI) that characterizes processing efficiency according to the temporal properties of stimulation. For instance, [30] were able to represent variations of visual processing efficiency according either to the temporal dimension alone or as a function of a time-frequency representation of the temporal samples (i.e. frequency spectrum of signal-to-noise ratio oscillations). Moreover, they showed that the power spectra of these CIs (extracted by Fourier transform) could be successfully used by a machine learning algorithm to map these patterns of temporal features onto the particular class of stimuli participants had to recognize. Specifically, the four-way mapping of individual patterns of temporal features to the task of recognizing words, familiar objects, unfamiliar objects, or faces was performed by the algorithm with an accuracy of 75%, which is far above the 25% chance level. In another study, [32] were able to predict whether participants were young adults or healthy elderly individuals with an accuracy above 90% based on individual data patterns extracted from classification images obtained in tasks of visual word or object recognition.

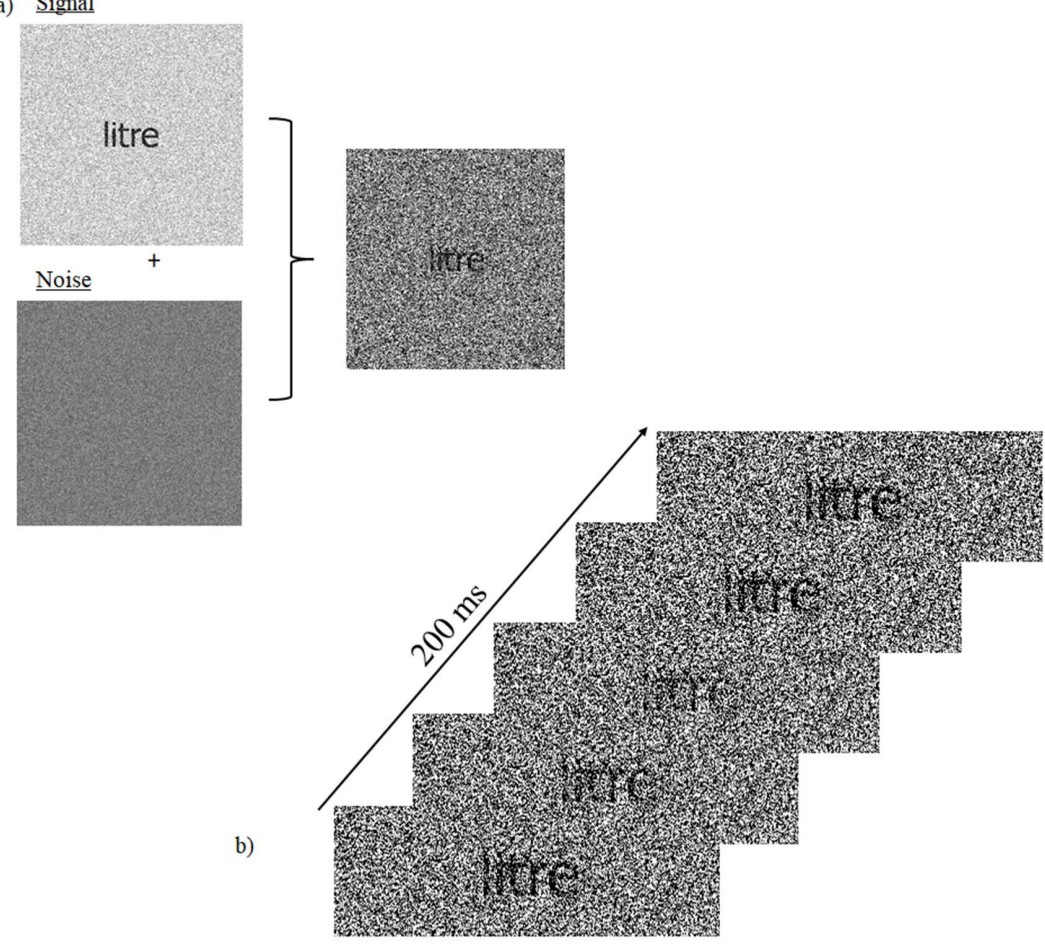

**Fig 1. Illustrations of the stimuli and the task.** Section a) shows the signal and noise components which are additively combined in each stimulus. Section b) gives an example of the time course of the stimulation in a given trial. Each trial was made of 24 successive displays of the additive combination of signal and noise. The signal/noise ratio varied following a random function integrating sine waves during display.

The highly likely origin of the temporal profiles of visual processing are the oscillatory mechanisms of the brain. Knowing that ADHD is associated with brain oscillations abnormalities at rest and during visuocognitive tasks, we should thus expect concomitant alterations in the temporal features of visual processing efficiency. The purpose of the present study is to compare the temporal features of visual word processing efficiency in young adults with vs without ADHD using the temporal sampling technique. As shown by the results below, there are marked differences between the temporal features of processing efficiency between ADHD and matched neurotypical controls. Moreover, these temporal features appear to be largely shared among individuals of the same group.

## Materials and methods

### Participants

Fifty-eight francophone participants were initially recruited from two colleges (pre-university or professional formations) in Montreal and Laval, Canada, via e-mails sent to students through the colleges' online platforms. The age of participants ranged from 16 to 35 years old. All had normal or corrected to normal vision and were free of neurological or psychiatric

disorders. Participants were separated into two groups. One was made of neurotypical controls, who reported normal function and never felt the need to consult about a possible ADHD. The other group was made of participants who reported having previously received an ADHD diagnosis from a qualified professional (medical doctor, psychiatrist, neuropsychologist or psychologist; i.e. the professions recognized for the emission of an ADHD diagnosis in Québec, Canada). The data of 49 of the 58 recruited participants was used for analysis. Of the 9 other participants, the data of one participant was rejected since they revealed, after data collection had started with them, that they had never actually received a formal diagnosis for ADHD. Another participant (neurotypical) was rejected because they had uncorrected vision problems, but only said so after data collection had started. Another seven participants (2 neurotypical controls and 5 ADHD) were removed from data analysis because they failed to complete the experiment or had missing data. The final sample was thus made of 26 neurotypical controls and 23 ADHD participants. Among the latter, 17 took stimulant medication on a regular basis for their condition and 6 did not. All participants provided written informed consent before enrolment. Consent was also needed from a parent or guardian for participants under 18 years of age to enroll in the study. Data collection took place between January 17, 2021 and October 19, 2021. The study was approved by the Comité d'Éthique de la Recherche en Éducation et Psychologie (Education and Psychology Research Ethics Committee) of Université de Montréal and the Comité d'Éthique de la Recherche (Research Ethics Committee) of Montmorency College prior to any participant recruitment.

### Materials and stimuli

Before experimentation, participants completed the Conners Adult ADHD Rating Scale – Long Version (CAARS; [33] as well as a general information form collecting descriptive variables such as age, gender, diagnosis and medication use.

The experiment was conducted on a HPZ230 computer with an NVIDIA GeForce GTX970 graphics card. Stimuli were presented on an Asus VG248QR HD monitor with a 120 Hz refresh rate. Stimuli were all achromatic and the luminance manipulations were linear. The experiment was programmed in Matlab and used the Psychophysics Toolbox [34]. Participants were seated in front of the screen, their head positioned on a chin rest 57 cm from the center of the screen.

Stimuli were 600 five-letter French words with an average frequency of 157 per 10 million [35]. Words were presented on screen for 200ms in Tahoma font with an x-height of.76 degrees of visual angle. Letters were black and the background was grey, with a luminance in the middle of the available range for the screen (between 1 and 195 Lux).

Stimuli were made of two components: signal and noise (Fig 1. a). The signal part consisted in the target word over which a patch of visual white noise was applied. The white noise patch was changed on every trial and its contrast was adjusted according to the procedure described below in order to maintain performance at around 50% correct. The noise component of the stimuli was made of a second visual white noise patch with maximal contrast, independent from the one that is part of the signal. This second white noise field also changed on every trial. The signal/noise ratio (SNR) varied throughout exposure duration following a random function constructed by integrating sine waves with frequencies ranging from 5 Hz to 55 Hz in steps of 5 Hz, with random amplitudes and phases (Fig 1. b). The SNR range was normalised between 0 and 0.5 and the sum of sampling functions across their constituent 24 values (1 per screen frame, which totals an exposure duration of 200ms at a refresh rate of 120 Hz), which represents the total stimulus availability for a trial, was constant across trials. The overall luminance and contrast of the stimuli were also matched across image frames and across trials.

### Procedure

Each participant completed 1200 trials (8 blocks of 150 trials each) wherein each of the 600 words was presented twice. Participants were asked to name aloud the words on the screen, without time pressure. As they read the words out loud, the experimenter entered the response on a computer keyboard. The program determined if the response was correct or incorrect and, if necessary, adjusted the contrast of the white noise that is part of the signal to control task difficulty (see below).

Each trial started with the presentation of a square white noise field (18 degrees of visual angle per side) at the center of the screen for 1250 ms. Then, a fixation cross was displayed for 250 ms at the center of the screen. Following a delay of 150 ms after the offset of the fixation cross, a pure tone of 900 Hz-60 dB was presented for 14 ms, indicating the imminent target onset. The use of this tone at that point in the course of a trial rests on the demonstration by [36] that it causes a reset of the neural oscillations in the human visual cortex. One hundred ms later, the target stimulus was presented at the center of the screen. During this 200 ms display, the SNR varied following a random function, as described above. The target display was followed by the white noise field with which the trial had begun, and the participant's response was entered on the keyboard by the experimenter.

The contrast of the white noise patch that was part of the signal was adjusted on each trial following a staircase function with 128 levels in order to maintain performance at about 50%. At the beginning of the experiment, the contrast was set at 64 and remained so for at least the first 10 trials. Following the 10th trial, accuracy for the previous 10 trials was assessed on every trial. If this accuracy was greater than 50%, white noise contrast was increased by one step. The reverse was done if accuracy was under 50%. The size of the initial step was 16 and it was halved on each reversal of contrast adjustment down to a minimum of 1. The state of the algorithm was maintained across consecutive experimental blocks. The total duration of the experiment was of about 2 hours, and it was completed in two test sessions occurring on different days, with breaks in-between each 15 minutes blocks.

## Data analysis

**Classification images.** Classification images (CIs) were calculated to depict how processing efficiency varied according to the features of the temporal sampling functions. Here, we focus on the classification images based on a time-frequency representation of the sampling functions, which revealed to be the most informative. To achieve this, a wavelet analysis was applied to the padded sampling functions on each trial using three-cycle complex Morlet wavelets. Padding was added to the beginning and end of the target SNR sampling function to be analyzed to avoid edge artifacts. This padding was made of 1.5 successive reversals of the sampling function connected end-to-end [37]. This way, the function submitted to analysis was continuous and signal was present along the entire length of even the lowest frequency wavelet when positioned at either end of the target SNR function. The time-frequency data retained from the analysis exclusively pertained to the target SNR function. These wavelets varied in temporal frequency from 5 to 55 Hz in increments of 5 Hz. The choice of the number of cycles in the Morlet kernel favored high temporal precision, albeit at the expense of precision in the frequency domain. Consequently, the wavelet exhibited sensitivity not only to its specific temporal frequency, but also to a range of frequencies around it.

Classification images were calculated for each participant. The weighted sum of the time-frequency sampling functions associated with errors was subtracted from the weighted sum of those associated with correct responses. These initial raw classification images were transformed in Z scores by a bootstrapping operation where the sampling functions were randomly assigned to response accuracies while allowing for repetition, and from which mock classification images were constructed. The mean and standard deviation of 1000 such mock classification images for an individual participant served as reference to transform the values from their raw classification image into Z scores.

The Z-scored individual classification images were averaged, smoothed, and then submitted to a two-way Pixel test [38] with α = .05 to determine the points in classification images which differed significantly from zero. The Pixel test is derived from random field theory and has been applied for about the last 30 years for the analysis of brain imaging data. Its purpose is to establish the Z value that will serve as the significance criterion for a Z-scored image. The smoothing filter was Gaussian and had a full width at half maximum (FWHM) of 19.6 ms in the time domain and of 11.8 Hz frequency domain. The criterion Z score obtained was then used in its positive value to identify points that were significantly above 0 and in its negative value (i.e., $Z_{crit} * -1$) to identify points significantly below 0.

A between-group-contrast CI was also calculated to compare the CIs of the ADHD vs neurotypical participants. Thus, the mean CI for the ADHD group was subtracted from that of the neurotypical group and the three preceding steps were repeated on the difference CI (bootstrap, smoothing, and Pixel test). The resulting CI thus showed the points of significant difference between the ADHD and neurotypical participants. A similar procedure was followed to contrast the CIs from the ADHD participants taking medication or not for their condition.

## Classification of individual data patterns

A further step of data processing was to submit features from CIs of individual participant to a machine learning algorithm for it to determine whether they came from a neurotypical or ADHD participant. The algorithm used for this purpose was a linear support vector machine (SVM; [39,40]), along with a leave-one-out cross-validation procedure. Thus, a subset of features from all but one of the available CIs were presented to the SVM for it to learn the mapping from these features to group. Then, the CI that had been left out of the learning phase was presented to the SVM for it to decide the group (neurotypical vs ADHD) it came from. This procedure was repeated by leaving out the data from a different participant on each iteration until it had iterated through the complete set of participants. Classification accuracy was determined from the percentage of iterations on which the SVM determined correctly the group of participants the data came from. A binomial test was used to assess whether classification accuracy deviated significantly from chance.

The features used for data pattern classification were those produced by the Fourier transform of the individual time-frequency CIs. Specifically, for each stimulus oscillation frequency in these CIs, a fast Fourier transform was applied to the variations of processing efficiency through time for frequencies between 5 and 60 Hz, in 5 Hz steps. This analysis produced a 3D feature space made of 1584 cells representing oscillatory power, with dimensions of frequency in the CI (12 levels), phase of the extracted components (binned in 12 levels), and the frequency spectrum of stimulus oscillation frequencies (11 levels). This particular data format was chosen because previous experience has shown that it offers the greatest discriminatory power to the classifier with a substantial increase of between-subject consistency and of the discrimination index of its features compared to the CIs themselves [31,32,41].

The classification of data patterns using an SVM satisfied several important aims. The most obvious is that an accuracy that is greater than chance implies that there exist significant relevant differences in the data patterns that are contrasted. Less obvious but crucially important is that it also provides an indication that these data patterns are replicable across individuals. Indeed, even if average data patterns are markedly different across the conditions compared, if they are not replicable across individuals, the performance of the classifier will be poor. In other words, to obtain a highly accurate classifier, the relevant features in the training sets must retain their value in the test pattern. Finally, another interesting aspect of using a classifier is that we can determine the features in the data patterns from which its discriminatory power is derived. This enables the characterization of the feature values that define each group.

In order to retain only the most relevant features that discriminate among conditions, we used a stepwise procedure for the introduction of features into the model one at a time, in a way similar to a stepwise multiple regression. This gradual introduction of features was pursued either until all of them were used or until 90% classification accuracy (to avoid overfitting) was reached. The order in which the features of classification images were introduced to the SVM model was based on the capacity of each possible feature to discriminate the ADHD and neurotypical groups. This discrimination capacity was analogous to an F ratio; i.e. it was measured by the ratio of the variance of the means across conditions over the error variance. Thus, the feature with the greatest discrimination index was entered first, followed by the second greatest, and so on, until the stopping criterion was reached.

For the illustration of the characteristic features of each level of a factor, the only data retained was that pertaining to the features used at the point where the stopping criterion was reached. The representation of a feature for each group was based on the squared difference between its mean and the overall mean across groups, which was divided by the error variance (see above). These values were then linearly normalized in the range -1 to 1 based upon the maximum absolute value among the

features to illustrate. To facilitate focussing on the strongest levers for classification, i.e., the features with the most extreme values, the contrast of the color code used to illustrate feature values was linearly diminished according to their distance from the extremes of the scale (i.e. -1 or 1), down to a minimum of 30% (to maintain visibility of even the weakest features illustrated). However, when the value of a feature for a particular condition was exactly 0, it was omitted from the figures.

A complementary data classification procedure was conducted using methods as described above to examine whether there are differences between the temporal features of visual processing in ADHD participants taking medication or not for their condition.

## Results

### Sample description

Groups were matched in terms of gender (22 (85%) women in the neurotypical control group and 16 (70%) women in the ADHD group, ($\chi^2$ (48) = 1.59; $p$ = .208; Cramer's V = .180) and age (overall mean = 19, s.d. = 2.8; F(1, 48) =.001; $p$ = .88; $\eta_p^2$ = .001). Participants with ADHD reported significantly more symptoms of inattention (average inattention score for the ADHD group: 63.91, s.d. = 12.11; average inattention for the neurotypical group: 55.24; s.d. = 9.7; F(1, 47) = 7.59; $p$ = .008; $\eta_p^2$ = .142) and hyperactivity (average hyperactivity score for the ADHD group: 55.5, s.d. = 9.44; average hyperactivity for the neurotypical group: 49.92, s.d. = 9.11; F(1, 47) = 4.32; $p$ = .043; $\eta_p^2$ = .086) on the CAARS.

The average correct response rate on the task was of 49.4% for the neurotypical group and 50.0% for the ADHD group ($t$(47) = −.92; $p$ = .181; Cohen's $d$ = −.27). The mean contrast of the white noise field applied over the target images, which served to control task difficulty was of 55.2% for neurotypical participants and 55% for ADHD participants ($t$(47) =.09; $p$ = .463; Cohen's $d$ = .03)

### ADHD vs neurotypical controls analyses

Group CIs are shown in Fig 2 for the neurotypical and ADHD groups, respectively. These CIs represent the capacity of participants to use the stimulus information available at each time point and for each SNR frequency (5 to 55 Hz) in order to reach a correct response; i.e. their processing efficiency. Cells colored in yellow and red indicate combinations of time and SNR frequencies where processing efficiency was significantly above 0, and blue cells indicate processing efficiency significantly below 0.

While the group CIs seem roughly similar, the CI for the contrast between groups (Fig 3) shows several statistically significant differences. The cells colored orange and red in Fig 3 indicate significantly greater efficiency for the neurotypical group whereas the blue cells indicate an advantage for the ADHD participants.

When exposed to the Fourier transforms of the individual time-frequency CIs, the SVM classifier reached an accuracy of 91.8% (binomial test; $p$ < 0.001) in classifying data patterns according to group while using only 51 (3.2%) of the 1584 features available. Inspection of the confusion matrix revealed a classification sensitivity of 96.2% and a specificity of 87%. The features used by the classifier for this performance are shown in Fig 4. The classifier managed to reach perfect (i.e. 100% correct) classification performance while using 421 features (not illustrated here).

The features which were most useful to distinguish between neurotypical and ADHD individual were the oscillations of processing efficiency (i.e. oscillations in the CIs) at 5, 10 and 15 Hz, which comprised the highest number of useful features, which were strongest for the 30, 35 and 40 Hz SNR oscillation frequencies. Across the 11 possible SNR oscillation frequencies, the 5 and 15 Hz stimulus frequencies were not or almost not used by the SVM, and the 10, 30, 35, 40 and 55 Hz frequencies were slightly more frequently used than the others. Of all features, though, three of them are stronger in their discriminatory index: the processing efficiency oscillations at 10 Hz for SNR oscillatory frequencies of 30 and 35 Hz, and the processing efficiency oscillations at 50 Hz for SNR oscillations at 20 Hz. Altogether, a higher concentration of informative features used by the SVM are found in the lower ranges of the frequencies in CIs (5, 10 and 15 Hz), but there is no clear pattern in which SNR oscillating frequencies are more used.

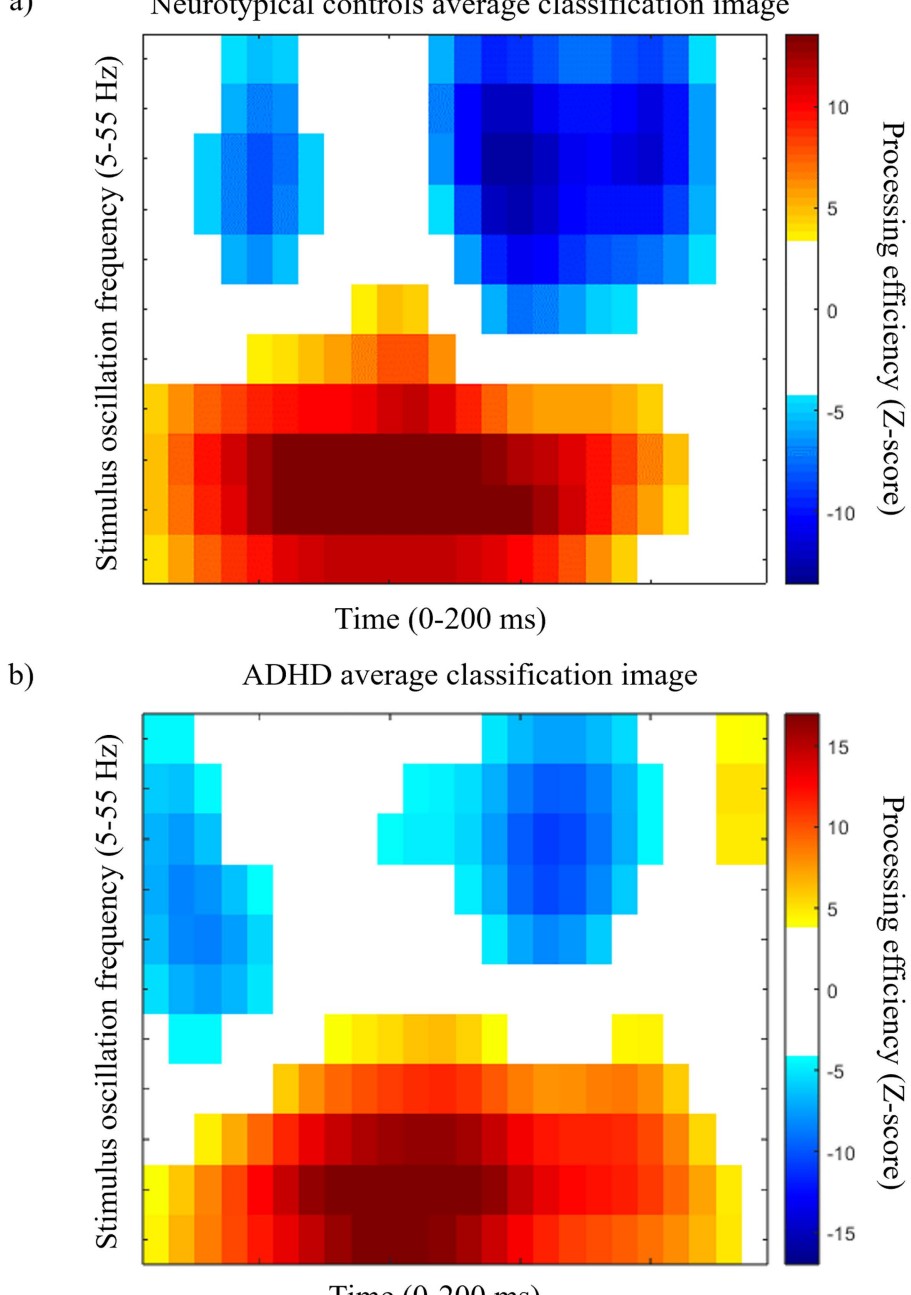

**Fig 2. ADHD and neurotypical participants' classification images.** Average classification images representing processing efficiency as a function of time and SNR oscillation frequencies for the neurotypical (a) and ADHD participants (b). Reference for the color code is on the right of each graph. Only the points that differ significantly from 0 are colored, the others are white.

## Medicated vs non-medicated ADHD participants analyses

The ADHD group was separated in two subgroups; one for participants who have prescription medication for their condition and who take it on a regular basis, and those who do not take medication. Neurotypical participants were not included in the remaining of the analyses. Of the 23 participants with ADHD, 17 were using medication and 6 were not.

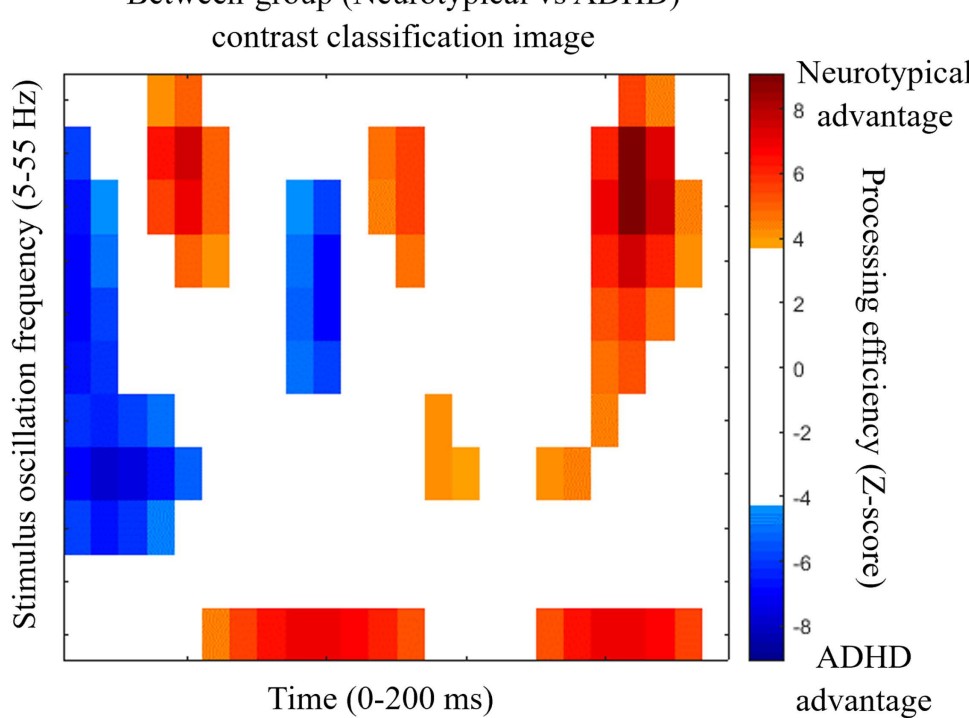

**Fig 3. Between groups contrast classification image.** The CI was obtained by subtracting the ADHD average CI from that of the neurotypical average CI. Conventions for the main axes of the graph are as in Fig 2. The color code represents the magnitude of between-group differences when significant. Cells that do not differ significantly between groups are white.

Subgroups were matched in terms of gender (4 (67%) women in the non-medicated group and 12 (71%) women in the medicated group ($\chi^2$ (23) =.03; $p$=.858; Cramer's V=.04), and age (overall mean=19.09, s.d.=3.68; F(1, 22) =.004; $p$=.948; $\eta_p^2$=.00). Participants also reported equivalent severity of inattentive symptoms on the CAARS inattention/memory symptoms subscale (overall mean=63.91, s.d.=12.11; F(1, 21) =.19; $p$=.662; $\eta_p^2$=.01), and hyperactive symptoms subscale (overall mean=55.5, s.d.=9.44; F(1, 21) =.09; $p$=.769; $\eta_p^2$=.00). The average correct response rate was of 50.3% for the non-medicated group and 49.9% for the medicated group ($t$(21) =.56; $p$=.289; Cohen's $d$=.27). The mean contrast of the white noise field applied over the target images (to control difficulty) was of 55.1% for non medicated participants and 54.9 for medicated participants (t(21) =.04; $p$=.49; Cohen's $d$=.08).

Classification images for the subgroups of ADHD participants taking medication for their condition on a regular basis or not were also calculated (Fig 5). The contrast CI for the comparison between these subgroups showed no significant difference and thus, is not illustrated here.

As in the case of the classification of ADHD vs neurotypical participants, an SVM was used to predict whether participants with ADHD take medication or not based on features from the Fourier transforms of individual time-frequency ICs. The SVM achieved a 91.3% decoding accuracy (binomial test; $p<0.001$) using only 8 (0.5%) of the 1584 available features (11 SNR oscillation frequencies x 12 phases x 12 CI dimensions=1584). Inspection of the confusion matrix revealed a classification sensitivity of 100% and a specificity of 66.7%. Fig 6. Illustrates the features used to reach this accuracy. Perfect decoding, with 100% correct classification, was reached based on 32 features (not illustrated here).

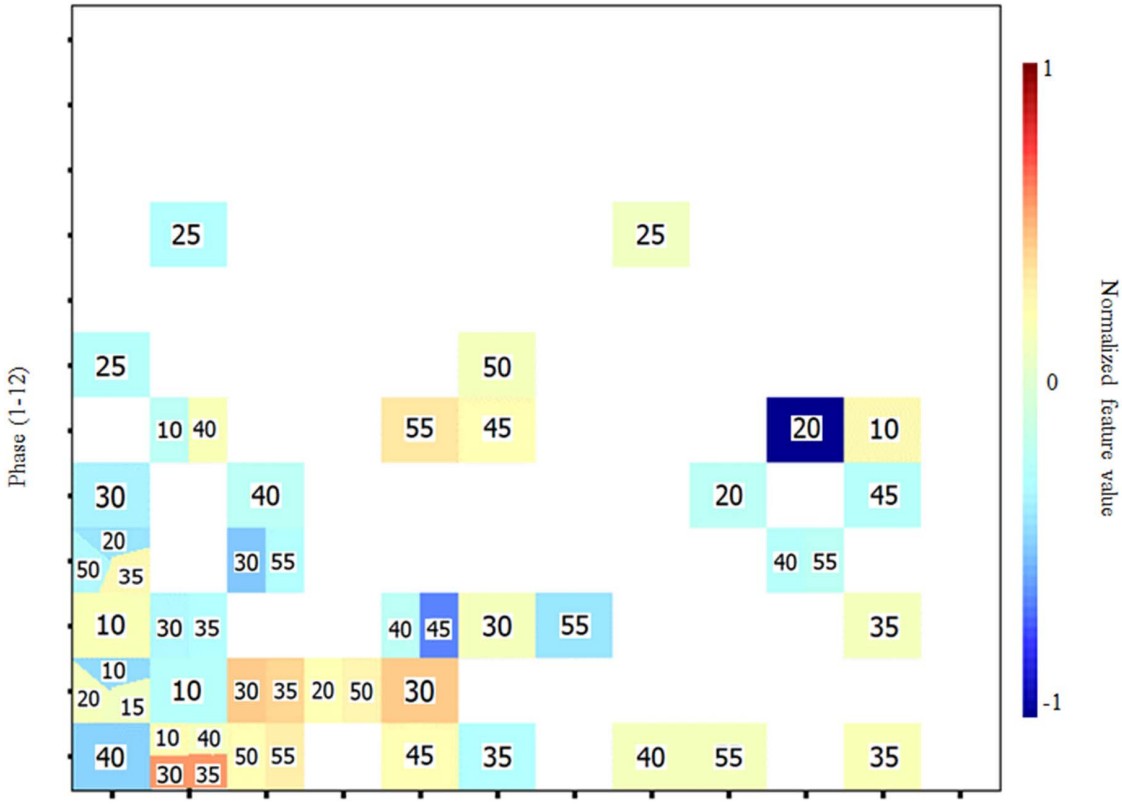

**Fig 4. Features used for diagnosis classification.** Characteristic features for the ADHD group which served for the SVM classifier to reach 91.8% accuracy in determining the group from which individual data patterns came from. The color of cells indicates the normalized feature discrimination index (see color bar legend). The horizontal axis indicates the temporal frequencies extracted from the CIs by Fourier analysis. The vertical axis indicates the phase of these extracted components. The digit within each colored cell indicates the stimulus oscillation frequency from which the feature was extracted. All cells left white did not contribute to classification. The characteristic features for the neurotypical control group can be obtained by simply multiplying the feature values shown here by −1.

A very limited number of features were used to discriminate between participants who take medication vs those who do not, and these features all had strong discrimination indexes. Although they are distributed across the possible SNR oscillating frequencies and CI frequencies, the strongest ones were oscillations in processing efficiency at 30 Hz for 20 Hz SNR oscillations, and processing efficiency oscillations at 20 Hz for 25 and 30 Hz SNR oscillations. Contrary to the ADHD vs control participants discrimination, in this case, low frequencies in processing efficiency variations were not very useful to distinguish between participants with ADHD who take medication vs those who do not.

## Discussion

The present study investigated the temporal fluctuations of visual processing efficiency in young adults with vs without ADHD using random temporal sampling. The average time-frequency CIs for both groups appeared rather similar and showed significant variations in processing efficiency both as a function of the time elapsed since target onset and as a function of the frequency spectrum of SNR oscillations in the stimulus. Rapid changes of processing capacity in the

a)

Medicated ADHD subgroup average
classification image

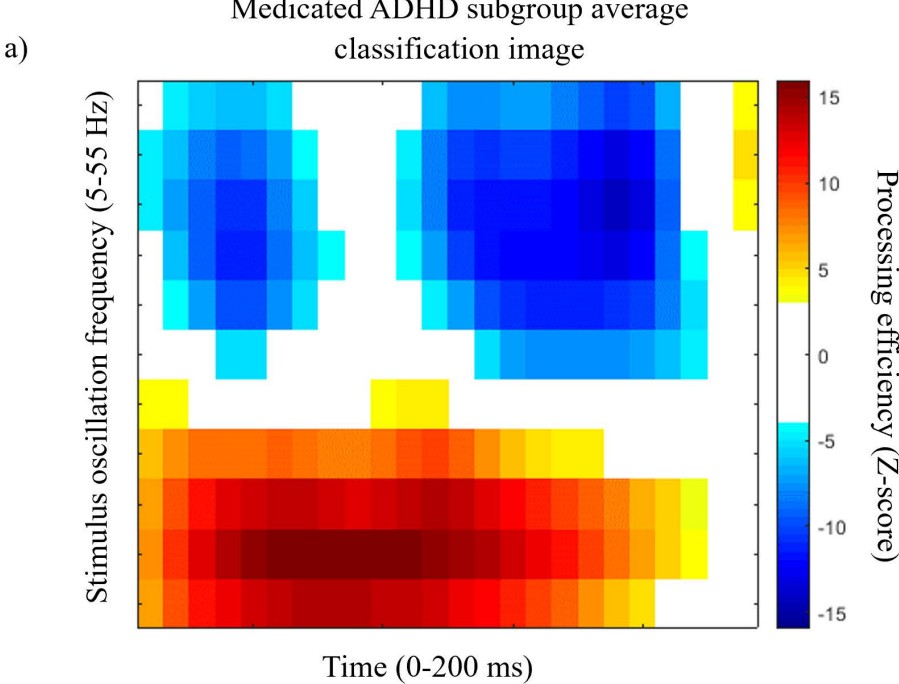

Time (0-200 ms)

b)

Non-medicated ADHD subgroup average
classification image

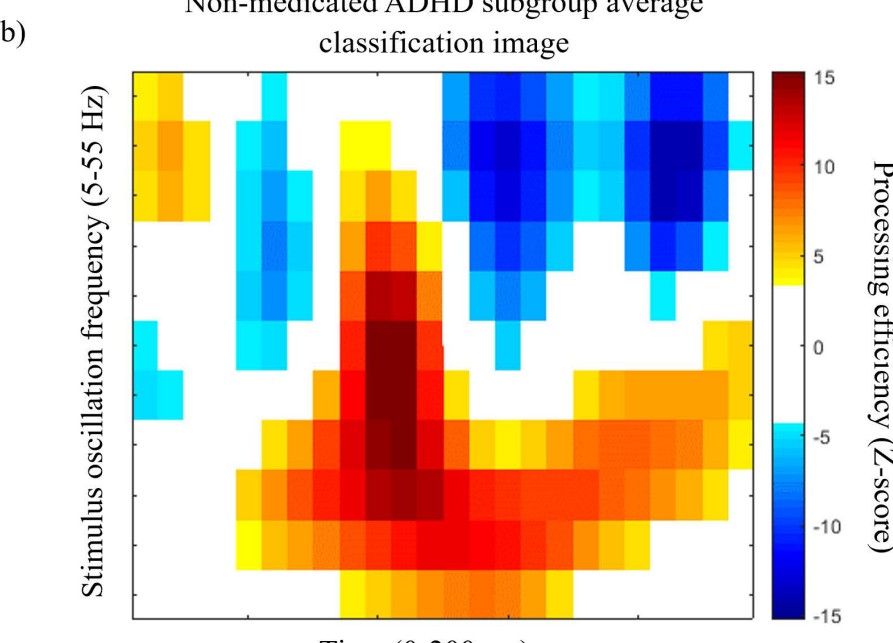

Time (0-200 ms)

**Fig 5. Medicated and non-medicated ADHD participants' classification images.** Average classification image representing processing efficiency as a function of time and SNR oscillation frequencies for the ADHD participants who take stimulant medication (a) and who do not take stimulant medication (b). Conventions are the same as in Fig 2.

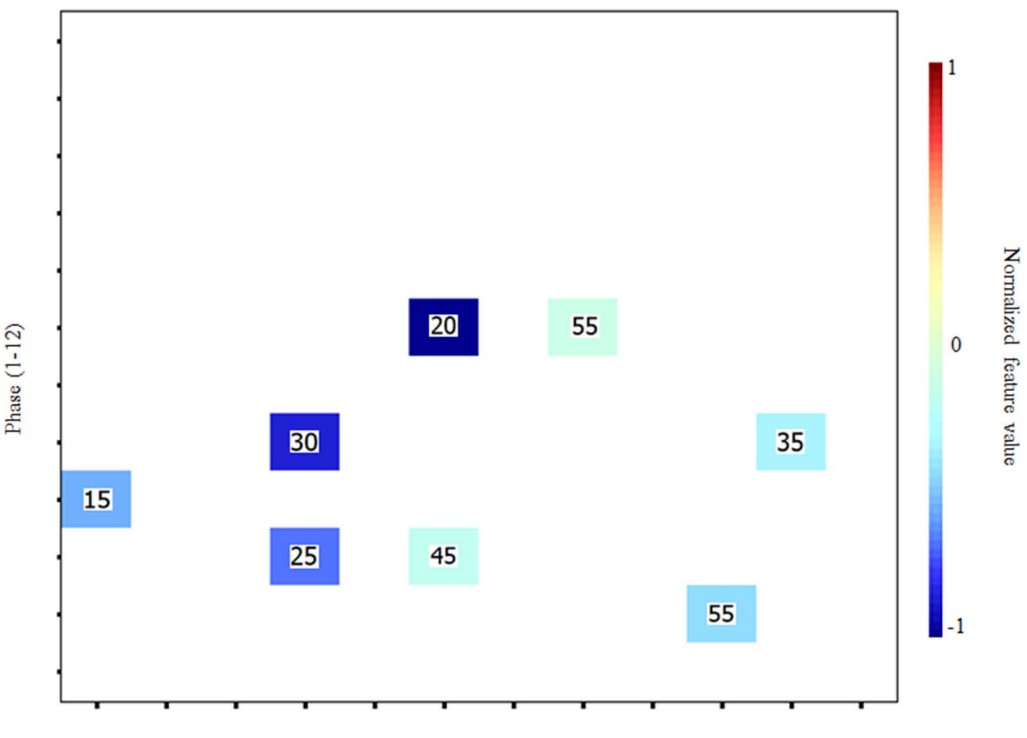

Characteristic features of the medicated ADHD subgroup

**Fig 6. Features used for medication status classification.** Characteristic features for the subgroup of ADHD participants who take medication on a regular basis for their condition. Conventions are the same as those for Fig 4.

course of a 200 ms stimulus exposure such as those illustrated in Figs 2 are most likely attributable to the neural oscillatory mechanisms underlying task performance. Specifically, such fluctuations imply that the system mediating the relation between stimuli and responses presents a form of temporal inhomogeneity in the timescale of 200 ms. Given current knowledge, we believe the best candidate to account for this is that of neural oscillations.

Significant differences were found between the average CIs of ADHD and neurotypical groups (Fig 3). These were confirmed through the use of an SVM classifier which had the task of categorizing features from the Fourier transforms of the CIs of individual participants according to their group of origin. Thus, the classifier surpassed the performance criterion of over 90% correct while using only 51 (3.2%) features out of the 3-D feature space of frequency within the classification image (12 levels; 5–60 Hz in 5 Hz steps) x phase (binned according to 12 levels) x stimulus oscillation frequency (11 levels; 5–55 Hz in 5 Hs steps) and reached a high level of sensitivity (96.2%) and specificity (87%). Furthermore, the classifier obtained a perfect performance of 100% classification accuracy while using 421 (26.6%) features to do so. The most salient features that served to discriminate between groups (Fig 4) largely pertain to low (5, 10 and 15 Hz) frequencies within the classification images, with a particular emphasis on 10 Hz, which comprised two of the three most discriminant features.

Considering that the temporal features of processing efficiency revealed by the classification images are a reflection of the neural oscillatory mechanisms underlying task performance, the group differences demonstrated here must be interpreted in terms of a significant alteration of these brain oscillations in the ADHD group. This conclusion is congruent with that of several EEG studies which compared the brain oscillatory activity of individuals with ADHD to that of neurotypical

participants. The fact that 5, 10 and 15 (to some extent) Hz processing efficiency oscillations were the best indicators of the presence or absence of ADHD among the possible frequencies is also consistent with the specific patterns of EEG oscillations alterations in ADHD reported across literature [15,42]. One significant constraint in interpreting the present observations, however, is that the way in which the temporal features of processing efficiency maps to specific neural mechanisms is presently unknown. This should be an important goal of future studies. While the issue must remain on standby at present, we underline that this limitation is well compensated by the remarkable capacity of the random temporal sampling technique of discriminating between cases of ADHD and neurotypical participants on the basis of a small number of features extracted from the data.

This high discrimination power is also showcased in the classification of ADHD participants who take stimulant medication for their condition on a regular basis versus those that do not. Thus, this classification task was achieved with a very high degree of accuracy (91.3%) and excellent sensitivity (100%) and relatively good specificity (66.7%) while using very few (n = 8; 0.5%) of the potential 1584 features available in the Fourier transforms of that time-frequency classification images. The lower specificity of the classification performance seems attributable to the small number of non-medicated participants, which is likely to have prevented the classifier to learn data patterns specifically associated with not using medication. A number of different factors may be involved in this difference between the medicated and non-medicated groups. One is that the classification images may have picked up the functional impact of the brain alterations caused by the long-term daily psychostimulant medication intake [43–45]. Thus, according to [46], the chronic use of psychostimulant medication in ADHD particularly impacts performance in tasks that are rather repetitive or that require little cognitive flexibility. The task used in the present study obeys both criteria, which may have made it highly susceptible to a medication effect. Concluding on this at the present time appears premature however, considering that the literature on the impacts of long-term medication use in ADHD is highly inconsistent [46,47]. An alternative account for the medication effect reported here is that the medicated and non-medicated ADHD participants actually suffer different degrees of symptom severity [48] and that this is the cause of the differences between their classification images. Specifically, it seems possible that severe ADHD is more likely to lead to the prescription/usage of medication than milder cases. A weak link in the latter hypothesis, however, is that self-reported symptoms were equivalent between medicated and non-medicated ADHD participants. This observation suggests that symptoms severity may not constitute the main difference between these groups. Regardless of its cause however, the present result clearly shows that brain function differs in some way between medicated and unmedicated individuals with ADHD. The existence of this difference means that future research focussing on cognitive and neural processes in ADHD should take the medication factor into account.

Another issue that needs to be addressed is that while an obvious difference in the brain's oscillatory activity was demonstrated between medicated and non-medicated participants through the classification of the Fourier transforms of time-frequency CIs, these CIs themselves failed to show a significant effect of medication. This very large difference between the outcomes of the two analysis techniques to contrast the data patterns from groups of participants is consistent with our past experience in the processing of data from random temporal sampling experiments. Specifically, we have found previously that both the discriminatory power of features as well as the between-subject consistency in the values of those feature are increased when using the features produced by the Fourier transform of time-frequency CIs rather than those coming from the CIs themselves [41,49,50].

For both the ADHD vs control and medication vs no-medication classification problems, this is partially verified here based on the discrimination index which determined the entry order of features in the SVM classifier and the intra-group correlation coefficient (ICC; [51]) to measure the consistency of data patterns across participants of the same group. As regards the ICC, it was lower with the Fourier transforms of CIs when compared to the CIs themselves in all cases except for the ADHD group in the contrast of their classification images with that of the control group (Table 1). However, large gains in the discrimination value of the features used by the SVM were evident with the Fourier transformed CIs (Table 1). Thus, the Fourier transform of CIs increased the discrimination value of features 1.9 times (i.e. nearly doubled) for the

**Table 1. Discrimination values of Classification Images features and Intra-class Correlations.**

| Controls vs ADHD | Feature discrimination index | Neurotypical control group | | | ADHD group | | |
|---|---|---|---|---|---|---|---|
| | | ICC | F(df) | p | ICC | F(df) | p |
| Time-Frequency CIs | .186 | .085 | 5.74 (9, 107) | < 0.001 | .041 | 2.78 (9, 145) | < 0.001 |
| Fourier transforms of T-F CIs | .357 | .043 | 2.17 (50, 1272) | < 0.001 | .067 | 2.65 (50, 1121) | < 0.001 |
| Medicated vs non-medicated | Feature discrimination index | Medicated ADHD subgroup | | | Non-medicated ADHD subgroup | | |
| | | ICC | F(df) | p | ICC | F(df) | p |
| Time-Frequency CIs | .476 | .169 | 5.4 (12, 159) | < 0.001 | .364 | 6.09 (12, 36) | < 0.001 |
| Fourier transforms of T-F CIs | 1.476 | −.039 | .4 (7, 81) | .900 | −.197 | .1 (7, 10) | 1.00 |

ADHD vs neurotypical classification problem, whereas the ratio was of 3.1 for classification of ADHD participants using medication vs not. We attribute this difference to an improved alignment (or correlation) of the data with the brain activity it reflects when the time dimension of the time-frequency CIs is recoded into a frequency spectrum. This position is consistent with the notion that the functional output of the brain is based on oscillatory neural activity. Specifically, a phase x amplitude frequency spectrum offers greater validity to characterize an oscillator than the time dimension.

Discrimination indexes of CI features and intra-group correlation coefficients for the time-frequency classification images and for the Fourier transforms thereof for the control vs ADHD group, and for the medicated vs non-medicated ADHD participants.

ADHD is known to be a very heterogeneous disorder [52,53]. However, the high accuracy of the SVM's classification points to high intra-group coherence in the data patterns. Indeed, the SVM classifications performed here used a leave-one-out cross validation method. This method implies that on every cycle, the data from one participant did not contribute to the learning of the mappings between data patterns and group while it is precisely the data that was left out from the learning phase that served for the test phase. To have an SVM classifier that offers a high accuracy, as in the present study, it is necessary that the mappings learned from the data of all participants but one retain their validity when the data of the left-out participant is presented to the classifier in the test phase. From this, we may thus conclude that there is an essence in the individual data patterns that is largely shared among other members of the same group. These findings point to temporal sampling as a promising method which, combined with machine learning, could help identify more homogeneous characteristics of ADHD and potentially be used as a powerful tool to assist in the diagnosis of ADHD and treatment monitoring. As regards the diagnosis issue, we have shown that there are specific features in the data patterns emerging from the task which are highly characteristic of ADHD (Fig 3). Thus, in the clinical context of cognitive screening, individuals exhibiting these features to a sufficiently high degree would appear as likely candidates for an ADHD diagnosis. Conversely, the lack of such features might prompt the consideration of alternative diagnoses. As regards to treatment monitoring, it is relevant to consider the similarity of the data patterns produced by the medicated vs non-medicated ADHD subgroups (Figs 5a and 5b) to that of the control participants (Fig 2a). It is quite obvious that the classification image of the medicated ADHD participants is much more similar to that of the controls than that of the non-medicated ADHD participants. This suggests that the use of medication may "normalize" the data patterns of ADHD individuals and, by extension, the brain activity underlying task execution. This "normalization" (or lack thereof) of the data patterns obtained with the random temporal sampling technique might therefore stand as an objective and useful candidate for the clinical assessment of whether a pharmacological treatment is effective.

Previous studies using random temporal sampling have shown that the data patterns emanating from the technique are very highly sensitive to task demands, stimulation parameters, and personal characteristics. For instance, markedly distinct classification images have been reported according to the class of stimulus that was to be recognized by young adults with normal cognitive function (i.e. visual words, faces, and familiar and unfamiliar objects; [30]). In a similar

sample, [31] have reported highly contrasting classification images according to the spatial frequency content of written stimuli. Furthermore, cognitively intact young adults and elderly persons have demonstrated highly contrasting data patterns in tasks of visual word or object recognition. However, the set of temporal features characterizing the elderly group was very different across tasks [32]. What such observations suggest is that the exact form of the contrast distinguishing ADHD participants from neurotypical individuals would most likely be strongly affected by factors such as stimulus class, cognitive load, or task demands. If one were to use the technique of random temporal sampling for diagnostic purposes then, it would be crucial to determine beforehand how ADHD participants differ from neurotypicals under the precise task and stimulation conditions that are to be used in the diagnostic test. In addition, special efforts should be made to make the method more accessible to clinicians who are not trained in the use of such techniques. Creating a "ready to use" program in which data collection and analysis are automated would be highly useful.

To push these questions further, future studies should include larger and more diverse populations in order to enhance external validity. Medication type, dosage and duration should also be measured and included in the analyses to assess their effects on neural oscillations. In addition, complementary neuroimaging studies would prove useful to relate the perceptual oscillatory patterns to neuronal oscillations occurring in specific brain regions and networks.

## Conclusion

The study compared the temporal features of visual processing between ADHD and neurotypical individuals in a word recognition task. These features were sufficiently different across groups while at the same time sufficiently congruent across participants of the same group that a machine learning algorithm classified participants in their respective groups with a 91.8% accuracy using only a small portion of the available features. This clearly shows that while ADHD is a very heterogeneous disorder [52,53], it remains possible to capture a highly powerful set of temporal features of visual processing that uniquely characterizes ADHD. Secondary findings showed that individuals with ADHD could be classified with high accuracy (91.3%) regarding their use of psychostimulant medication. This thus suggests the existence of strong behavioral markers of regular medication usage on visual performance which can be uncovered by random temporal sampling.

## Author contributions

**Conceptualization:** Martin Arguin.

**Data curation:** Pénélope Pelland-Goulet, Martin Arguin.

**Formal analysis:** Pénélope Pelland-Goulet, Martin Arguin.

**Funding acquisition:** Hélène Brisebois, Nathalie Gosselin.

**Investigation:** Pénélope Pelland-Goulet.

**Methodology:** Pénélope Pelland-Goulet, Martin Arguin.

**Project administration:** Pénélope Pelland-Goulet, Martin Arguin, Hélène Brisebois, Nathalie Gosselin.

**Resources:** Pénélope Pelland-Goulet, Hélène Brisebois, Nathalie Gosselin.

**Software:** Martin Arguin.

**Supervision:** Martin Arguin, Hélène Brisebois, Nathalie Gosselin.

**Validation:** Pénélope Pelland-Goulet, Martin Arguin.

**Visualization:** Pénélope Pelland-Goulet, Martin Arguin.

**Writing – original draft:** Pénélope Pelland-Goulet, Martin Arguin.

**Writing – review & editing:** Pénélope Pelland-Goulet, Martin Arguin, Hélène Brisebois, Nathalie Gosselin.

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
