## [Decision Letter · Decision Letter 0]

26 Nov 2024

PONE-D-24-35136Visual processing oscillates differently through time for adults with ADHDPLOS ONE

Dear Dr. Pelland-Goulet,

Thank you for submitting your manuscript to PLOS ONE. After careful consideration, we feel that it has merit but does not fully meet PLOS ONE’s publication criteria as it currently stands. Therefore, we invite you to submit a revised version of the manuscript that addresses the points raised during the review process.

We look forward to receiving your revised manuscript.

Kind regards,

Tsai-Ching Hsu, Ph.D.

Academic Editor

PLOS ONE

Journal Requirements:

Additional Editor Comments:

The authors need some detailed explanation of the medical assessments section and add more discussion sections as the reviewers suggested.

Reviewers' comments:

Reviewer's Responses to Questions

**Comments to the Author**

1. Is the manuscript technically sound, and do the data support the conclusions?

Reviewer #1: Yes

Reviewer #2: Yes

2. Has the statistical analysis been performed appropriately and rigorously? 

Reviewer #1: Yes

Reviewer #2: Yes

3. Have the authors made all data underlying the findings in their manuscript fully available?

Reviewer #1: Yes

Reviewer #2: Yes

4. Is the manuscript presented in an intelligible fashion and written in standard English?

Reviewer #1: Yes

Reviewer #2: Yes

5. Review Comments to the Author

Reviewer #1: Technical Soundness and Rigor:

"The manuscript demonstrates high technical rigor, with each step of the methodology carried out exactly as outlined. The authors have implemented each procedure consistently and with clear adherence to established standards in the field."

"The analytical methods are appropriately selected and well-executed, ensuring accurate interpretation of the findings."

Data Alignment with Conclusions:

"The conclusions are directly supported by the data, with clear evidence presented for each claim. Each result is systematically connected to the corresponding data, making the conclusions reliable and well-founded."

"The sample size and data quality lend strong credibility to the study, and the statistical analyses applied are appropriate, further substantiating the authors’ conclusions."

Additional Minor Comments (if any):

"Although the manuscript meets high standards of technical precision, minor clarifications, such as an expanded explanation of [specific section], may further enhance clarity for readers."

Reviewer #2: Strengths

This study creatively combines random temporal sampling techniques and machine learning algorithms to explore differences in visual processing efficiency between individuals with ADHD and neurotypical controls. This approach not only reveals potential neural oscillatory features but also demonstrates its clinical application potential.

The high classification accuracy of machine learning (91.8%) highlights the effectiveness of the research methodology and provides strong evidence supporting the feasibility of oscillatory features as biomarkers for ADHD.

The study conducts an in-depth comparison of ADHD individuals and further analyzes differences between medicated and non-medicated participants, providing a solid foundation for future research on medication effects.

By integrating techniques and knowledge from neuroscience, psychology, and computer science, this research holds significant interdisciplinary value and is likely to attract attention from researchers across multiple fields.

Limitations and Suggestions for Improvement

The sample size is relatively small (49 participants), particularly the group of non-medicated ADHD participants, which only includes 6 individuals. This limitation may affect the generalizability of the results. It is recommended to expand the sample size in future studies to enhance the external validity of the findings.

The study does not specify the types and dosages of stimulant medications used by participants, which could significantly influence the results. Future research should collect and analyze these details to assess the differential impacts of various medications on neural oscillations.

Were ADHD diagnoses based on medical assessments? Were participants diagnosed in early childhood? Clarifying these aspects is crucial to understanding the heterogeneity within the study population.

While the diagnostic value of oscillatory features is demonstrated, the study lacks a thorough exploration of their association with specific neural mechanisms. It is suggested that future research further investigate the relationship between these features and the functions of specific brain regions.

The article provides limited discussion on how oscillatory features could be translated into concrete clinical tools. Adding practical examples or potential application scenarios in clinical diagnostics and treatment monitoring would enhance the article's applicability.

Conclusion and Overall Evaluation

This study demonstrates the potential of random temporal sampling techniques and machine learning in identifying neural oscillatory features associated with ADHD. It holds significant scientific value and clinical applicability. However, limitations such as the small sample size and insufficient details on stimulant usage restrict the generalizability of the conclusions.

The innovative study design and comprehensive data analysis lay a solid foundation for future research. I recommend the study for publication following appropriate revisions to the discussion section, and I look forward to seeing further studies that validate and expand upon these findings.

6. PLOS authors have the option to publish the peer review history of their article (what does this mean? ). If published, this will include your full peer review and any attached files.

**Do you want your identity to be public for this peer review?** For information about this choice, including consent withdrawal, please see our Privacy Policy .

Reviewer #1: No

Reviewer #2: No

---

## [Author Response · Author response to Decision Letter 1]

19 Dec 2024

We have examined the comments, questions and suggestions which arose from the review of the manuscript. In what follows, we will address the issues which were raised. Each change made to the manuscript is noted below along with its page number.

We hope that you will find our responses and manuscript changes satisfactory.

Best regards,

The authors

Pénélope Pelland-Goulet, Martin Arguin, Hélène Brisebois & Nathalie Gosselin

General editor suggestions

Q1. Please ensure that your manuscript meets PLOS ONE’s style requirements, including those for file naming. The PLOS ONE style templates can be found at

A1. We have thoroughly checked the style/formatting requirements and that the present resubmission is in accordance. Slight formatting errors in affiliations were corrected (p. 1; lines 7-15).

We have also made minor corrections to figure 1 (enhanced visibility corrected the lowercase first word letter (“Litre” became “litre”) and to figures 3, 4, and 6’s titles. To favour format uniformity across all figures, we reupdated them all.

Q2. Please include your full ethics statement in the ‘Methods’ section of your manuscript file. In your statement, please include the full name of the IRB or ethics committee who approved or waived your study, as well as whether or not you obtained informed written or verbal. If consent was waived for your study, please include this information in your statement as well.

A2. The full ethics statement may be found on p. 5 (lines 99-103).

“Consent was also needed from a parent or guardian for participants under 18 years of age to enroll in the study. Data collection took place between January 17, 2021 and October 19, 2021. The study was approved by the Comité d’Éthique de la Recherche en Éducation et Psychologie (Education and Psychology Research Ethics Committee) of Université de Montréal and the Comité d'Éthique de la Recherche (Research Ethics Committee) of Montmorency College prior to any participant recruitment.”

Q3. Please review your reference list to ensure that it is complete and correct. If you have cited papers that have been retracted, please include the rationale for doing so in the manuscript text, or remove these references and replace them with relevant current references. Any changes to the reference list should be mentioned in the rebuttal letter that accompanies your revised manuscript. If you need to cite a retracted article, indicate the article’s retracted status in the Reference list and also include a citation and full reference for the retraction notice.

A3. We have done so. We also searched each article cited in our manuscript on the Retraction Watch Database (https://retractiondatabase.org/RetractionSearch.aspx) and found no match. for any of the references. Finally, we checked whether errata/corrections have been published for each cited article and found none.

Finally, please note that we have also added the following citations in the revised manuscript (all checked as above).

Methods:

p. 7; line 144 : Romei V, Gross J, Thut G. Sounds reset rhythms of visual cortex and corresponding human visual perception. Curr Biol. 2012 May 8;22(9):807-13. doi: 10.1016/j.cub.2012.03.025. Epub 2012 Apr 12. PMID: 22503499; PMCID: PMC3368263.

Discussion:

p. 16 & 17; lines 381 & 385 : Swanson J, Baler RD, Volkow ND. Understanding the effects of stimulant medications on cognition in individuals with attention-deficit hyperactivity disorder: a decade of progress. Neuropsychopharmacology. 2011 Jan;36(1):207-26. doi: 10.1038/npp.2010.160. Epub 2010 Sep 29. PMID: 20881946; PMCID: PMC3055506.

p. 17; line 385 : Bidwell LC, McClernon FJ, Kollins SH. Cognitive enhancers for the treatment of ADHD. Pharmacol Biochem Behav. 2011 Aug;99(2):262-74. doi: 10.1016/j.pbb.2011.05.002. Epub 2011 May 10. PMID: 21596055; PMCID: PMC3353150.

In p. 17; line 405, have also removed one reference (Milanova, Singh & Arguin, in preparation, Université de Montréal (previously #46)) and replaced it with the following references:

Milanova G & Arguin M. The processing sequence in visual object recognition investigated by temporal sampling. Communication. Annual conference of the Société Québécoise pour la Recherche en Psychologie (SQRP). 25 mai 2024, Drummondville, Québec, Canada.

Bertrand Pilon C & Arguin M. Le traitement des fréquences spatiales à travers le temps dans la reconnaissance des mots écrits. Communication. Annual conference of the Société Québécoise pour la Recherche en Psychologie (SQRP). 25 mai 2024, Drummondville, Québec, Canada.

Q4. Additional editor comments: The authors need some detailed explanation of the medical assessments section and add more discussion sections as the reviewers suggested.

A4. More information regarding medical assessment and discussion sections have been added. These are described in detail in our replies to reviewers.

Reviewer #1

Technical Soundness and Rigor:

Q1. The manuscript demonstrates high technical rigor, with each step of the methodology carried out exactly as outlined. The authors have implemented each procedure consistently and with clear adherence to established standards in the field.

Q2. The analytical methods are appropriately selected and well-executed, ensuring accurate interpretation of the findings.

Data Alignment with Conclusions:

Q3. The conclusions are directly supported by the data, with clear evidence presented for each claim. Each result is systematically connected to the corresponding data, making the conclusions reliable and well-founded.

Q4. The sample size and data quality lend strong credibility to the study, and the statistical analyses applied are appropriate, further substantiating the authors’ conclusions.

A1-4. We appreciate the favorable evaluation of our methods.

Additional Minor Comments (if any):

Q5. Although the manuscript meets high standards of technical precision, minor clarifications, such as an expanded explanation of [specific section], may further enhance clarity for readers.

A5. Any methodological detail that we found was missing from the initial manuscript has been added in the present revision (see p. 5; line 116, p. 7; lines 144-146). Slight reformulations were also made to improve clarity in this section (see p. 6; lines 136, 138, p. 7; lines 147, 152 & 156, p. 9; lines 192, 193 & 199, p. 9; line 200, p. 11; line 245).

Reviewer #2

Strengths:

Q1. This study creatively combines random temporal sampling techniques and machine learning algorithms to explore differences in visual processing efficiency between individuals with ADHD and neurotypical controls. This approach not only reveals potential neural oscillatory features but also demonstrates its clinical application potential.

Q2. The high classification accuracy of machine learning (91.8%) highlights the effectiveness of the research methodology and provides strong evidence supporting the feasibility of oscillatory features as biomarkers for ADHD.

Q3. The study conducts an in-depth comparison of ADHD individuals and further analyzes differences between medicated and non-medicated participants, providing a solid foundation for future research on medication effects.

Q4. By integrating techniques and knowledge from neuroscience, psychology, and computer science, this research holds significant interdisciplinary value and is likely to attract attention from researchers across multiple fields.

A1-4. Thank you for these positive comments on the manuscript and methods.

Limitations and Suggestions for Improvement:

Q5. The sample size is relatively small (49 participants), particularly the group of non-medicated ADHD participants, which only includes 6 individuals. This limitation may affect the generalizability of the results. It is recommended to expand the sample size in future studies to enhance the external validity of the findings.

A5. It is true that the sample size may seem limited. However, this sample size is relatively large for studies of this kind (e.g., 10 participants in Wang et al., 2011; 41 and 59 participants for two experiments in Blais et al., 2012; 19 participants in Wutz, Melcher & Samaha, 2018).

Moreover, as noted in the manuscript, the classification of data patterns according to the presence/absence of ADHD or the use of medication or not in ADHD participants along with a leave-one-out cross validation procedure constitutes a fairly robust support for the generalizability of our findings. Specifically, on every iteration, the classifier had to learn a correspondence pattern between the data patterns and the classification criteria for all participants but one, to then be tested on the data pattern that had been left out from the learning phase. Thus, to be successful, the mapping learned by the classifier had to remain valid for the test pattern for it to be successful For this reason, we maintain that the high classification rates obtained (up to 100%) offer in this study suggest that the discriminant information identified by the classifier for the “ADHD vs control” and the “medication vs not” problems is quite generalizable.

Wang HF, Friel N, Gosselin F, Schyns PG. Efficient bubbles for visual categorization tasks. Vision Res. 2011 Jun 21;51(12):1318-23. doi: 10.1016/j.visres.2011.04.007. Epub 2011 Apr 16. PMID: 21524660.

Blais C, Roy C, Fiset D, Arguin M, Gosselin F. The eyes are not the window to basic emotions. Neuropsychologia. 2012 Oct;50(12):2830-2838. DOI: 10.1016/j.neuropsychologia.2012.08.010. PMID: 22974675.

Wutz A, Melcher D, Samaha J. Frequency modulation of neural oscillations according to visual task demands. Proc Natl Acad Sci U S A. 2018 Feb 6;115(6):1346-1351. doi: 10.1073/pnas.1713318115. Epub 2018 Jan 22. PMID: 29358390; PMCID: PMC5819398.

Q6. The study does not specify the types and dosages of stimulant medications used by participants, which could significantly influence the results. Future research should collect and analyze these details to assess the differential impacts of various medications on neural oscillations.

A6. It is true that the exact type of medication and dosage used would be relevant information. Unfortunately, we did not ask the questions relevant to this issue in our General information questionnaire. We will make sure we do so in future studies. We should nevertheless emphasize that even without medication type/dosage information, it was possible to classify ADHD participants according to their use of medication or not with high accuracy. We take this to suggest that we were able to capture a common alteration of brain function that is exerted by the medications used in our ADHD sample, regardless of the exact medication type/dosage. This issue is further addressed on p. 16-17 (lines 379-386).

Relevant to the topic of medication are also lines 439-446 (p. 19-20) which were added in response to Question #9.

Q7. Were ADHD diagnoses based on medical assessments? Were participants diagnosed in early childhood? Clarifying these aspects is crucial to understanding the heterogeneity within the study population.

A7. We agree that these informations are relevant.

We do have information as to the profession of the persons who emitted the ADHD diagnosis and made sure that they are from professions abilitated to do so. We added those professions to the sample description. We address the issue in the revised manuscript on p. 4 (lines 88-90).

Unfortunately, however, we have no information as to the age of diagnosis.

Q8. While the diagnostic value of oscillatory features is demonstrated, the study lacks a thorough exploration of their association with specific neural mechanisms. It is suggested that future research further investigate the relationship between these features and the functions of specific brain regions.

A8. We absolutely agree with this view and work is underway in our lab to study the way the perceptual oscillations demonstrated by the technique of random temporal sampling relate to brain activity.

We have made more explicit the need for explicit demonstrations of the relationship between perceptual oscillations and brain activity (see p. 16; lines 368-369).

Q9. The article provides limited discussion on how oscillatory features could be translated into concrete clinical tools. Adding practical examples or potential application scenarios in clinical diagnostics and treatment monitoring would enhance the article's applicability.

A9. We have added discussion of this topic on p. 19-20 (lines 434-446).

---

## [Decision Letter · Decision Letter 1]

29 Jan 2025

PONE-D-24-35136R1Visual processing oscillates differently through time for adults with ADHDPLOS ONE Dear Dr. Pelland-Goulet,

Thank you for submitting your manuscript to PLOS ONE. After careful consideration, we feel that it has merit but does not fully meet PLOS ONE’s publication criteria as it currently stands. Therefore, we invite you to submit a revised version of the manuscript that addresses the points raised during the review process.

We look forward to receiving your revised manuscript.

Kind regards,

Tsai-Ching Hsu, Ph.D.

Academic Editor

PLOS ONE

Journal Requirements:

Reviewers' comments:

Reviewer's Responses to Questions

**Comments to the Author**

1. If the authors have adequately addressed your comments raised in a previous round of review and you feel that this manuscript is now acceptable for publication, you may indicate that here to bypass the “Comments to the Author” section, enter your conflict of interest statement in the “Confidential to Editor” section, and submit your "Accept" recommendation.

Reviewer #2: All comments have been addressed

Reviewer #3: (No Response)

2. Is the manuscript technically sound, and do the data support the conclusions?

Reviewer #2: Yes

Reviewer #3: Yes

3. Has the statistical analysis been performed appropriately and rigorously? 

Reviewer #2: Yes

Reviewer #3: Yes

4. Have the authors made all data underlying the findings in their manuscript fully available?

Reviewer #2: Yes

Reviewer #3: Yes

5. Is the manuscript presented in an intelligible fashion and written in standard English?

Reviewer #2: Yes

Reviewer #3: Yes

6. Review Comments to the Author

Reviewer #2: Suggestions for Improvement:

Expand Sample Size: Future studies should include larger and more diverse populations to enhance external validity.

Collect Detailed Medication Data: Include information on medication types, dosages, and duration to better assess their effects on neural oscillations.

Explore Neural Mechanisms: Conduct complementary neuroimaging studies to link the observed oscillatory patterns to specific brain regions and networks.

Address Comorbidities: Account for the presence of comorbid psychiatric or neurological conditions to isolate ADHD-specific effects.

Additional Comments:

The manuscript is well-structured, with clear figures and thorough methodological details. However, a more extensive discussion of the practical challenges in translating oscillatory features into clinical tools would enhance its applicability.

Consider discussing the potential impact of task demands and cognitive load on the observed oscillatory patterns.

The manuscript provides a strong foundation for understanding visual processing differences in ADHD and demonstrates the potential of random temporal sampling for diagnostic and monitoring purposes. Addressing the noted limitations would further strengthen its contribution to the field.

Reviewer #3: line 30 temporal sampling: Could you describe it more definitely

line 42/44 TBR ?? dose it mean theta/beta ratio??,please remark it clearly

in the section of material and method

can you mention the classification of neurotypical control , or it just the same type group of diagnosis?

if possible , what is the presentation of normal people group on temporal sampling and could you mention it in your article ?

line 402

for the medication vs none problem??

does it mean none stimulants medication? if it is, please modify it more clearly

7. PLOS authors have the option to publish the peer review history of their article (what does this mean? ). If published, this will include your full peer review and any attached files.

**Do you want your identity to be public for this peer review?** For information about this choice, including consent withdrawal, please see our Privacy Policy .

Reviewer #2: No

Reviewer #3: No

---

## [Author Response · Author response to Decision Letter 2]

6 Feb 2025

February 6, 2025

Editorial office

PLoS One

Object: Manuscript PONE-D-24-35136R1, by Pelland-Goulet et al.

We have examined the comments, questions and suggestions which arose from the latest review of the manuscript. In what follows, we will address the issues which were raised. Each change made to the manuscript is noted below along with its page number.

We hope that you will find our responses and manuscript changes satisfactory.

Best regards,

The authors

Pénélope Pelland-Goulet, Martin Arguin, Hélène Brisebois & Nathalie Gosselin

Reviewer #2

Suggestions for improvements

Q1. Expand Sample Size: Future studies should include larger and more diverse populations to enhance external validity. Collect Detailed Medication Data: Include information on medication types, dosages, and duration to better assess their effects on neural oscillations. Explore Neural Mechanisms: Conduct complementary neuroimaging studies to link the observed oscillatory patterns to specific brain regions and networks.

Thank you for these propositions. We have added the elements listed in your comment in the discussion (p. 20, lines 467-471).

Additional Comments

Q2. The manuscript is well-structured, with clear figures and thorough methodological details. However, a more extensive discussion of the practical challenges in translating oscillatory features into clinical tools would enhance its applicability.

Indeed, there are still challenges to overcome before the technique can be used in clinical settings. We have added discussion of this topic (p. 20, lines 464-466).

Q3. Consider discussing the potential impact of task demands and cognitive load on the observed oscillatory patterns. The manuscript provides a strong foundation for understanding visual processing differences in ADHD and demonstrates the potential of random temporal sampling for diagnostic and monitoring purposes. Addressing the noted limitations would further strengthen its contribution to the field.

We have added a more in-depth discussion of these topics (p. 20, lines 451-464).

Reviewer #3

Comments

Q1. “line 30 temporal sampling: Could you describe it more definitely”

We have added more details regarding the temporal sampling method (p. 2, lines 24-26).

Q2. “line 42/44 TBR ?? dose it mean theta/beta ratio??,please remark it clearly”

Thank you for bringing this to our attention. We have clarified the exact wording of the TBR abbreviation (p. 2, lines 43-44).

Q3. “in the section of material and method can you mention the classification of neurotypical control , or it just the same type group of diagnosis? if possible , what is the presentation of normal people group on temporal sampling and could you mention it in your article?”

We have clarified the way in which participants were assigned to either the neurotypical or ADHD group (p. 4, lines 89-90).

As for the description of neurotypical participants group, we already describe in detail their results in the manuscript and there is no important information that was left out. It is not clear from R3’s comment what should be added in this regard.

Q4. “line 402 for the medication vs none problem?? does it mean none stimulants medication? if it is, please modify it more clearly”

We have revised the paragraph to clarify what data is discussed in this section (p. 17, line 407; p. 18; lines 413-417).

---

## [Decision Letter · Decision Letter 2]

9 Apr 2025

PONE-D-24-35136R2Visual processing oscillates differently through time for adults with ADHDPLOS ONE

Dear Dr. Pelland-Goulet,

Thank you for submitting your manuscript to PLOS ONE. After careful consideration, we feel that it has merit but does not fully meet PLOS ONE’s publication criteria as it currently stands. Therefore, we invite you to submit a revised version of the manuscript that addresses the points raised during the review process.

Please submit your revised manuscript by May 24 2025 11:59PM, If you will need more time than this to complete your revisions, please reply to this message or contact the journal office at plosone@plos.org . Please include the following items when submitting your revised manuscript:

We look forward to receiving your revised manuscript.

Kind regards,

Tsai-Ching Hsu, Ph.D.

Academic Editor

PLOS ONE

Journal Requirements:

Reviewers' comments:

Reviewer's Responses to Questions

**Comments to the Author**

1. If the authors have adequately addressed your comments raised in a previous round of review and you feel that this manuscript is now acceptable for publication, you may indicate that here to bypass the “Comments to the Author” section, enter your conflict of interest statement in the “Confidential to Editor” section, and submit your "Accept" recommendation.

Reviewer #4: All comments have been addressed

Reviewer #5: All comments have been addressed

2. Is the manuscript technically sound, and do the data support the conclusions?

Reviewer #4: Partly

Reviewer #5: Yes

3. Has the statistical analysis been performed appropriately and rigorously? 

Reviewer #4: Yes

Reviewer #5: Yes

4. Have the authors made all data underlying the findings in their manuscript fully available?

Reviewer #4: Yes

Reviewer #5: Yes

5. Is the manuscript presented in an intelligible fashion and written in standard English?

Reviewer #4: Yes

Reviewer #5: Yes

6. Review Comments to the Author

Reviewer #4: The final sample included only 49 participants (26 neurotypical, 23 ADHD), which is very small, especially given the goal of distinguishing individuals based on subtle neurocognitive markers. This small size can lead to statistical noise being mistaken for true signal.

The machine learning classifier achieved 91.8% accuracy using a subset of features, which is impressive. However, there is no indication of external validation, such as applying the model to a new dataset or performing a stratified k-fold cross-validation.

P-values are used throughout for statistical comparisons (e.g., for CAARS scores, gender balance), but no effect sizes are provided.

Reviewer #5: Dear Authors,

Following the review of successive versions of this manuscript, it is noted that the work has adequately addressed the issues raised by the reviewers. This study represents a relevant contribution to the understanding of ADHD and to the field of cognitive and computational neuroscience. The work is situated at the intersection of perceptual psychophysics, computational neuroscience – particularly predictive processing models and oscillatory dynamics – and the emerging area of digital biomarkers for neurodevelopmental disorders. The adopted approach combines the random temporal sampling (RTS) methodology with machine learning (SVM) classification to investigate the temporal dynamics of visual perceptual processing, revealing aspects of brain function in adult ADHD that may not be captured by more static approaches.

Strengths and In-depth Theoretical Integration

1.RTS as a Tool to Investigate Predictive Dynamics: The RTS methodology, generating time-frequency CIs, functions as a psychophysical tool to investigate the temporal precision and efficiency of information utilization under uncertainty (noise). This aligns with contemporary theoretical frameworks that view the brain as a predictive inference system (e.g., Friston, 2005; Hohwy, 2013), constantly confronting internal predictions (priors) with afferent sensory evidence to compute prediction errors. Alterations in this process have been proposed in neurodevelopmental disorders (e.g., Gonzalez-Gadea et al., 2015; Samengo et al., 2022). The fluctuations in visual processing efficiency captured by the CIs in this study (Figures 2, 3) can be interpreted as a behavioral manifestation of the stability (or instability) with which the ADHD neural system generates, weights, and updates these predictions. The frequency-domain analysis (pp. 7-8) aligns with the view of neural oscillations as a substrate for timing, sensory gating, and inter-area communication (Buzsáki, 2006; Klimesch, 2012). The methodological rigor employed (pp. 7-10) confers robustness to the current findings.

2.The Temporal Phenotype of ADHD – Oscillations, Filtering, and Arousal: The high accuracy of the SVM classifier in distinguishing ADHD vs. NT (91.8%; p. 12) suggests a consistent and potentially replicable processing signature in ADHD (high intra-group coherence; p. 18), despite its clinical heterogeneity (Faraone et al., 2015; Luo et al., 2019). This signature may reflect systematic alterations in the parameters governing the weighting between predictions and sensory error, or in the efficacy of mechanisms for filtering irrelevant stimuli – a concept related to latent inhibition, a filtering process that may be altered in ADHD (e.g., Micoulaud-Franchi et al., 2015). The importance of low frequencies (5-15 Hz, alpha/theta; p. 13) as key discriminators is relevant, as alpha/theta rhythms are linked to inhibitory gating and anticipatory timing (Haigh & Buckby, 2024; Kannen et al., 2024; Klimesch, 2012). The observed pattern may indicate a deficit in these mechanisms, resulting in greater vulnerability to distraction. It can be hypothesized that such processing instability and low threshold for distraction connect to the need for movement, viewed as a compensatory strategy to regulate cortical arousal in ADHD (Hartanto et al., 2016; Sikström & Söderlund, 2007), possibly via modulation of ascending systems such as the ARAS.

3.Medication Effects as Neuromodulatory Recalibration: The ability to classify medication status (91.3%; p. 14) with a distinct set of spectral characteristics (Figure 6) is a relevant finding. It suggests that psychostimulants may recalibrate the predictive system, perhaps by strengthening priors or optimizing sensory gain control via DA/NA systems that modulate thalamo-cortical circuits and ascending activating systems (e.g., Mamiya et al., 2021; Sara & Bouret, 2012; Volkow et al., 2011). The apparent "normalization" of the CI pattern in medicated individuals (Figure 5 vs. Figure 2a) may represent the behavioral correlate of this optimization (p. 19), consistent with some findings on long-term effects of stimulants (Konrad et al., 2007; Swanson et al., 2011). The difference in discriminatory frequency profiles (p. 14) may indicate distinct targets: central instability versus adaptive gain/precision adjustments.

4.Convergence with Digital and Multimodal Assessment: This work contributes to the search for objective and digital markers in ADHD (Keshav et al., 2019; Teruel et al., 2024; Zhang et al., 2024). The RTS/SVM methodology employed here represents a sophisticated form of digital phenotyping. While methods such as AR/VR/games aim for ecological validity (Keshav et al., 2019; Teruel et al., 2024; Zheng et al., 2024; Kannen et al., 2024), the current paradigm offers experimental control to isolate temporal mechanisms. Convergence exists: performance in digital tasks correlates with clinical measures (Keshav et al., 2019; Teruel et al., 2024). The fusion of multimodal data improves classification (Zhang et al., 2024), suggesting that the visual findings presented here are part of a broader profile. The success of SVM in this study and of DNNs (Zhang et al., 2024) indicates the utility of machine learning. The sensitivity of RTS to SNR manipulation (p. 6) aligns with the importance of cognitive load in revealing deficits (Zhang et al., 2024; Teruel et al., 2024).

5.Alpha Oscillations, Modulation Complexity, and Heterogeneity: The relevance of the alpha band (p. 13) can be contrasted with the difficulty in directly modulating alpha power via tACS (Kannen et al., 2024), pointing to the complexity of oscillatory dynamics in ADHD. Processing efficiency linked to alpha may represent a more robust behavioral phenotype than neural power itself. Although the study finds high intra-group consistency (p. 18), the heterogeneity of ADHD (Faraone et al., 2015; Luo et al., 2019) and the variability in basic sensory processes (Jiang et al., 2024) indicate the need for future research on subtypes (Kannen et al., 2024). Conceptually, the difficulty in controlling attention, reflected in the oscillations, aligns with the idea that attentional bias influences perception (Rooney et al., 2024).

Evaluation

Technical Soundness and Data Support: The manuscript (R2) demonstrates technical soundness within its scope. The methods are appropriate, and the analyses robustly support the conclusions regarding the differential temporal signature in adult ADHD. The distinction between perceptual efficiency and neural mechanisms is made with scientific prudence (p. 16). Its value lies in providing quantitative empirical parameters for computational models of ADHD.

Limitations and Theoretical Breadth: The limitations (small non-medicated subgroup size, lack of medication details) are acknowledged and well-discussed (pp. 16-17, 20). They contextualize the inferences but do not diminish the importance of the central finding of the classifiable temporal signature. The discussion situates the findings within multiple relevant theoretical frameworks (oscillations, prediction, neuromodulation, digital assessment).

Conclusion

This manuscript presents an original investigation, utilizing a pertinent methodology with theoretical relevance regarding the temporal bases of visual processing in adult ADHD. The authors have successfully demonstrated that the oscillatory dynamics of perceptual efficiency, revealed by random temporal sampling and quantified by machine learning, constitute a robust signature that differentiates individuals with ADHD from controls and reflects medication status. These findings provide empirical support for models that conceive of ADHD not merely as a static attention deficit, but as an alteration in the dynamic regulation of information processing, potentially linked to an atypical balance between prediction and sensory error. The work contributes to the understanding of ADHD as a condition affecting the dynamic regulation of the brain's interaction with a constantly changing world. The conclusions are well-supported by the data, the limitations are addressed with scientific integrity, and the potential for future research – including clinical applications as a biomarker and transdiagnostic investigations – is evident. This study demonstrates how rigorous psychophysics, combined with advanced computational analyses and a deep neuroscientific foundation, can advance our understanding of complex disorders such as ADHD. The manuscript, in its current form (R2), meets the publication criteria for PLOS ONE. I recommend its acceptance.

7. PLOS authors have the option to publish the peer review history of their article (what does this mean? ). If published, this will include your full peer review and any attached files.

**Do you want your identity to be public for this peer review?** For information about this choice, including consent withdrawal, please see our Privacy Policy .

Reviewer #4: No

Reviewer #5: No

---

## [Author Response · Author response to Decision Letter 3]

12 May 2025

Reviewer #4

Review Comments to the Author

Q1. The final sample included only 49 participants (26 neurotypical, 23 ADHD), which is very small, especially given the goal of distinguishing individuals based on subtle neurocognitive markers. This small size can lead to statistical noise being mistaken for true signal.

A1. It is true that the sample size may seem limited. We also believe that for these markers to be used in clinical settings, the results would need to be replicated with different, larger samples, to ensure that the observations are truly generalizable across ADHD adults. However, this sample size is relatively large for studies investigating visual mechanisms using techniques similar to ours (e.g., 10 participants in Wang et al., 2011; 41 and 59 participants for two experiments in Blais et al., 2012; 19 participants in Wutz, Melcher & Samaha, 2018).

Moreover, as noted in the manuscript, the classification of data patterns according to the presence/absence of ADHD or the use of medication or not in ADHD participants along with a Leave-one-out cross validation procedure constitutes a fairly robust support for the generalizability of our findings. Specifically, on every iteration, the classifier had to learn the correspondence between the data patterns and the categories for classification for all participants but one, to then be tested on the data pattern that had been left out from the learning phase. Thus, to be successful, the mapping learned by the classifier had to remain valid for the test pattern for it to be successful. For this reason, we maintain that the high classification rates obtained (up to 100%) in this study suggest that the discriminant information identified by the classifier for the “ADHD vs control” and the “medication vs not” problems is rather generalizable.

Blais, C., Roy, C., Fiset, D., Arguin, M. & Gosselin, F. (2012). The eyes are not the window to basic emotions. Neuropsychologia, 50(12):2830-2838. DOI: 10.1016/j.neuropsychologia.2012.08.010.

Wang, H. F., Friel, N., Gosselin, F., Schyns, P. G. (2011). Efficient bubbles for visual categorization tasks. Vision Res, 51(12):1318-23. doi: 10.1016/j.visres.2011.04.007.

Wutz, A., Melcher, D. & Samaha, J. (2018) Frequency modulation of neural oscillations according to visual task demands. Proc Natl Acad Sci U S A, 115(6):1346-1351. doi: 10.1073/pnas.1713318115.

Q2. The machine learning classifier achieved 91.8% accuracy using a subset of features, which is impressive. However, there is no indication of external validation, such as applying the model to a new dataset or performing a stratified k-fold cross-validation.

A2. Regarding the method of cross-validation, it is true that we did not test the model on a brand-new dataset. However, we used a Leave-one-out (LOO) cross-validation method, which is capable of addressing much of the issues that can be addressed with stratified k-fold cross-validation (Yadav & Shukla, 2016; Sreedharan et al., 2023).

While the first classification problem compared groups of similar sample sizes (23 ADHD participants vs 26 neurotypical participants), it is true that the second classification problem compared unequal groups (6 non-medicated vs 17 medicated ADHD participants), which created an imbalance in the quantity of data representing each label in this classification task.

In the interest of ensuring maximal validity of our results, as per the suggestion of Reviewer 4, we tested the stratified k-fold cross-validation method (Prusty, Patnaik & Dash, 2022). However, the results were not satisfying: the classifier only reached 69.2% accuracy using all available data when predicting group (ADHD vs neurotypical) and 75.5% accuracy using all data when predicting medication usage among ADHD participants.

Our choice of the LOO method was made based on the observation that for small samples LOO is more appropriate, while k-fold is more appropriate for larger samples (Yadav & Shukla, 2016). Indeed, the LOO method ensures that a maximum of data is used during the training phase while testing the model on each individual participant when it was excluded from the training set (Sreedharan et al., 2023). In our case, even using a very small k (two folds) reduced the training set to rather small numbers, especially for the non-medicated group. These small samples seem to have been insufficient for the SVM to properly capture the crucial features by which groups differ. Indeed, according to Akbani, Kwek & Japkowicz (2004), undersampling the largest of the imbalanced samples (i.e. reducing its size to be more similar to that of the smallest sample) reduces the available information available for the model, limiting its accuracy.

Overall then, we maintain our use of the LOO cross-validation method, which appears as the most appropriate given the present sample. Data using the k-fold method is not reported given the sample size problem it runs into, as noted above.

In order to describe the classification results in greater detail, we have computed the sensitivity and specificity for both classifications problems at the stopping point of 90% or above classification performance. When participants were classified according to whether they have ADHD or not, we obtained a sensitivity of 96.2% and a specificity of 87%. For the medication classification, sensitivity was of 100%, with a specificity of 66.7%. These numbers are excellent, except for the specificity of the medication classification problem. The lower specificity of 66.7% for this problem may be attributed to the small number of non-medicated participants, which seems to have prevented the classifier to learn data patterns specifically associated with not using medication.

We have added the sensitivity and specificity indexes in the manuscript at p. 12, lines 282-283 (group classification), p. 14, lines 331-332 (medication classification) and p. 15, line 360; p. 16, lines 381-386 (general discussion).

Akbani, R., Kwek, S., Japkowicz, N. (2004). Applying Support Vector Machines to Imbalanced Datasets. In: Boulicaut, JF., Esposito, F., Giannotti, F., Pedreschi, D. (eds) Machine Learning: ECML 2004. ECML 2004. Lecture Notes in Computer Science, vol 3201. Springer, Berlin, Heidelberg. https://doi.org/10.1007/978-3-540-30115-8_7

Prusty, Sashikanta, Patnaik, Srikanta & Dash, Sujit Kumar. (2022). SKCV: Stratified K-fold cross-validation on ML classifiers for predicting cervical cancer. Frontiers in Nanotechnology, 4. https://doi.org/10.3389/fnano.2022.972421

Sreedharan, R., Prajapati, J., Engineer, P. & Prajapati, D. (2023). Leave-One-Out Cross-Validation in Machine Learning. Ethical Issues in AI for Bioinformatics and Chemoinformatics, 1st edition. pp. 56-71. http://dx.doi.org/10.1201/9781003353751-5

Yadav, S. & Shukla, S. (2016). Analysis of k-Fold Cross-Validation over Hold-Out Validation on Colossal Datasets for Quality Classification. 2016 IEEE 6th International Conference on Advanced Computing (IACC), Bhimavaram, India, pp. 78-83, doi: 10.1109/IACC.2016.25.

Q3. P-values are used throughout for statistical comparisons (e.g., for CAARS scores, gender balance), but no effect sizes are provided.

A3. Thank you for this observation. We have added the effect sizes for statistical comparisons (ADHD vs neurotypical comparisons (p. 11, lines 252 to 260) and ADHD medicated vs ADHD non-medicated (p. 13-14, lines 311-318)).

Reviewer #5

Review comments to the Author

The manuscript, in its current form (R2), meets the publication criteria for PLOS ONE. I recommend its acceptance.

We want to thank Reviewer 5 for taking the time to make such a thorough examination of our manuscript and providing in depth analysis of our results.

R5 asked for no additional modification to the manuscript.

---

## [Decision Letter · Decision Letter 3]

14 Aug 2025

Visual processing oscillates differently through time for adults with ADHD

PONE-D-24-35136R3

Dear Dr. Pénélope Pelland-Goulet,

We’re pleased to inform you that your manuscript has been judged scientifically suitable for publication and will be formally accepted for publication once it meets all outstanding technical requirements.

Kind regards,

Tsai-Ching Hsu, Ph.D.

Academic Editor

PLOS ONE

Additional Editor Comments (optional):

Reviewers' comments:

Reviewer's Responses to Questions

**Comments to the Author**

1. If the authors have adequately addressed your comments raised in a previous round of review and you feel that this manuscript is now acceptable for publication, you may indicate that here to bypass the “Comments to the Author” section, enter your conflict of interest statement in the “Confidential to Editor” section, and submit your "Accept" recommendation.

Reviewer #6: All comments have been addressed

Reviewer #7: All comments have been addressed

Reviewer #8: All comments have been addressed

2. Is the manuscript technically sound, and do the data support the conclusions?

Reviewer #6: Yes

Reviewer #7: Yes

Reviewer #8: Yes

3. Has the statistical analysis been performed appropriately and rigorously? 

Reviewer #6: Yes

Reviewer #7: Yes

Reviewer #8: Yes

4. Have the authors made all data underlying the findings in their manuscript fully available?

Reviewer #6: Yes

Reviewer #7: Yes

Reviewer #8: Yes

5. Is the manuscript presented in an intelligible fashion and written in standard English?

Reviewer #6: Yes

Reviewer #7: Yes

Reviewer #8: Yes

6. Review Comments to the Author

Reviewer #6: Using random temporal sampling combined with machine learning, this study identifies unique oscillatory patterns in visual processing that distinguish individuals with ADHD from neurotypical counterparts. Additionally, the authors suggested that these temporal features can predict psychostimulant medication use with high accuracy.

After reviewing the revised manuscript, I find that the article has undergone substantial improvement. It provides novel insights into the temporal dynamics of visual processing in adults with ADHD. The integration of random temporal sampling and machine learning represents an innovative approach, yielding high classification accuracy in distinguishing between clinical and control groups. Additionally, the study highlights promising applications in monitoring treatment adherence and personalizing interventions. I recommend that this article be accepted in its current form.

Reviewer #7: This manuscript presents a compelling study examining temporal features of visual processing in adults with ADHD using a novel temporal sampling technique and machine learning classification. The methodology is sound and innovative, and the findings, especially regarding the discriminative power of visual processing features and medication status, are both novel and potentially clinically relevant.

Reviewer #8: This paper presents an interesting study that applies machine learning to study temporal features of visual processing using a visual word recognition task and a temporal sampling technique. The study is, novel and timely. Methods are well described and clear. Results are interesting. This is a re-submission and most of the review comments seem to be addressed, except the ones about small population size and suggested replication in an independent dataset. The authors have responded to one previous reviewer "We also believe that for these markers to be used in clinical settings, the results would need to be replicated with different, larger samples, to ensure that the observations are truly generalizable across ADHD adults." THIS IS A VERY KEY FACT and should be highlighted appropriately in the manuscript. These kind of studies using novel techniques are very exciting and important and can pave the way for future innovations, but especially in clinical applications, where they may impact lives of real people, it is equally and especially important highlight caveats very prominently for any future reader for full context.

The sample is small as pointed out by previous reviewers. Moreover many subjects were dropped and the proportion of omitted subjects varies by group. It needs to be ensured this is not associated with a bias in an already very small sample. (7 participants - 2 neurotypical controls and 5 ADHD for failure to complete the experiment or missing data).

Furthermore, the age range is very large 16-35 and may overlap with developmental milestones. Significantly, ADHD is a neurodevelopmental disorder - several features of ADHD are believed to show a developmental lag, which transforms around 18 years of age in some individuals - for example, several brain regions such as the caudate are smaller in ADHD but the difference in volume reduces around 18 years. Thus we need to be sure what percent of subjects are less than 18 and that this does not interfere with the results.

Additionally, the recruitment and selection screening does not seem sound - according to my understanding of what was described. If Conners/CAARS questionnaire was used anyway in the study, then could this not be employed to screen ADHD participants, rather than go by self report? Validity of self-report is potentially less reliable in clinical groups.

Finally, they report using CAARS (adult version of Conners) but some of the participants are under 18 so should Conners not be used for those?

The study uses cross-validation to evaluate the model and achieves a very high accuracy. The standard Leave-One-Out Cross-Validation approach is applied, which makes sense given the small sample size. However, the study does not include a separate test set, which raises concerns about possible overfitting and limits confidence in how well the results would generalize to new data. A more robust design would have included a held-out test set. The method is creative, but the validation strategy could be improved to strengthen the conclusions.

Very minor points: Cover page has 2 typos in abstract: “Its symptoms include inattention, hyperactivity and impulsivity. Tough ” - should be though? And 3-4 of Canadian adults and 2.6% of adults worldwide - should be 3-4 %? + in line 44 vs. should be vs (for consistency)

7. PLOS authors have the option to publish the peer review history of their article (what does this mean? ). If published, this will include your full peer review and any attached files.

**Do you want your identity to be public for this peer review?** For information about this choice, including consent withdrawal, please see our Privacy Policy .

Reviewer #6: No

Reviewer #7: **Yes: ** Mykhailo Sosnov

Reviewer #8: No

---

## [Editor Report · Acceptance letter]

PONE-D-24-35136R3

PLOS ONE

Dear Dr. Pelland-Goulet,

I'm pleased to inform you that your manuscript has been deemed suitable for publication in PLOS ONE. Congratulations! Your manuscript is now being handed over to our production team.

Kind regards,

on behalf of

Dr. Tsai-Ching Hsu

Academic Editor

PLOS ONE